# Estimation of Above-Ground Carbon Storage and Light Saturation Value in Northeastern China's Natural Forests Using Different Spatial Regression Models

**Simin Wu** [1,2] , **Yuman Sun** [1,2] , **Weiwei Jia** [1,2,*] , **Fan Wang** [1,2] , **Shixin Lu** [1,2] **and Haiping Zhao** [1,2]

1   School of Forestry, Northeast Forestry University, Harbin 150040, China; wusimin1017@nefu.edu.cn (S.W.); symsfs@nefu.edu.cn (Y.S.); wangfan@nefu.edu.cn (F.W.); losbubble@nefu.edu.cn (S.L.); zhaohp2021@nefu.edu.cn (H.Z.)
2   Key Laboratory of Sustainable Forest Ecosystem Management, Ministry of Education, Northeast Forestry University, Harbin 150040, China
*   Correspondence: jiaww@nefu.edu.cn

**Abstract:** In recent years, accurate estimation and spatial mapping of above-ground carbon (AGC) storage in forests have been crucial for formulating carbon trading policies and promoting sustainable development strategies. Forest structure complexities mean that during their growth, trees may be affected by the surrounding environment, giving rise to spatial autocorrelation and heterogeneity in nearby forest segments. When estimating forest AGC through remote sensing, data saturation can arise in dense forest stands, adding to the uncertainties in AGC estimation. Our study used field-measured stand factors data from 138 forest fire risk plots located in Fenglin County in the Northeastern region, set within a series of temperate forest environments in 2021 and Sentinel-2 remote sensing image data with a spatial resolution of 10 m. Using ordinary least squares (OLS) as a baseline, we constructed and compared it against four spatial regression models, spatial lag model (SLM), spatial error model (SEM), spatial Durbin model (SDM), and geographically weighted regression (GWR), to better understand forest AGC spatial distribution. The results of local spatial analysis reveal significant spatial effects among plot data. The GWR model outperformed others with an $R^2$ value of 0.695 and the lowest rRMSE at 0.273, considering spatial heterogeneity and extending the threshold range for AGC estimation. To address the challenge of light saturation during AGC estimation, we deployed traditional linear functions, the generalized additive model (GAM), and the quantile generalized additive model (QGAM). AGC light saturation values derived from QGAM most accurately reflect the actual conditions, with the forests in Fenglin County exhibiting a light saturation range of 108.832 to 129.894 Mg/ha. The GWR effectively alleviated the impact of data saturation, thereby reducing the uncertainty of AGC spatial distribution in Fenglin County. Overall, accurate predictions of large-scale forest carbon storage provide valuable guidance for forest management, forest conservation, and the promotion of sustainable development strategies.

**Keywords:** above-ground carbon storage; spatial mapping; spatial autocorrelation and spatial heterogeneity; light saturation value; Sentinel-2 images; uncertainty

## 1. Introduction

Forests play a central and indispensable role in the global carbon cycle [1,2]. The forest ecosystem is characterized by species richness, structural complexity, and diverse resources [3]. The powerful carbon sequestration ability of forests plays a crucial role in climate change [4,5]. Forest above-ground carbon (AGC) storage is defined as the total amount of carbon stored in the above-ground components of a forest ecosystem, including tree trunks, branches, and leaves [6,7]. Forest AGC represents a more stable indicator of long-term carbon accumulation and is an essential attribute for reflecting the dynamics of forest ecosystems [8,9]. Although most studies indicate that estimating AGC in

forests comes with uncertainty [10–12], accurately assessing its spatial distribution remains essential for climate change mitigation and shaping carbon trading policies [13].

According to convention, forest AGC estimation methods can be categorized into field measurement techniques based on allometric equations [14,15], detailed biophysical models [16,17], and empirical models that combine field data with various remote sensing data, including optical, thermal, microwave, radar, and LiDAR data [18]. The advantage of remote sensing technology lies in its ability to effortlessly collect information on forest types and coverage, facilitating large-scale, long-term, and repetitive monitoring [19,20]. As an economically efficient approach, it is widely employed for the extensive estimation of forest AGC [21]. Multisensory data have been widely used for AGC mapping and are a primary data source for AGC estimation [22,23]. However, the processing and analysis of multisensory data can pose complexities due to the diverse characteristics and calibration requirements of different sensors [24]. Furthermore, cost and feasibility considerations need to be accounted for when acquiring data. LiDAR data, unaffected by lighting conditions, offer high-precision estimation by capturing canopy height information. It is considered to be a promising technology for AGC estimation [25,26]. However, due to its high cost, the complexity of forest structure, and the challenges associated with data processing, LiDAR data are predominantly employed in small-scale areas. Optical remote sensing data are a commonly used data source for estimating AGC [27,28]. Medium-resolution and high-resolution data are usually used for local-scale AGC estimation, such as Landsat series, SPOT, Sentinel series, and GF series [11,29–31]. Despite the significant limitations of optical imagery, including relatively low estimation accuracy, lack of consistency, and significant initial costs to acquire and produce results, it still serves as an alternative method for mapping large-scale forest AGC [27]. High-resolution imagery can be used to gather more detailed vegetation information, such as vegetation indices and texture features [32–34]. These data sets are typically used for parametric or nonparametric estimation methods by establishing a complete mathematical model and combining remote sensing image information with ground standard survey data. Ultimately, analysis formulas are used to estimate AGC [35]. Compared to parametric models, nonparametric models generally exhibit superior data fitting ability and higher estimation accuracy [36,37]. However, nonparametric models are more susceptible to the size and representativeness of the sample, meaning that their effectiveness may be limited in study areas with smaller sample sizes [38]. As a result, addressing the challenge of reducing uncertainty in estimating forest AGC based on remote sensing data remains a significant area of research [11,38,39].

Differences in AGC levels are anticipated across different regions due to variations in geographic location, site conditions, soil characteristics, and climate [40]. Vegetation types and their structures and physiologies are influenced to varying degrees by the surrounding environment during the growth process, which is manifested in forestry data as spatial correlations between adjacent trees. As trees compete, spatial heterogeneity becomes evident [41]. Although the spatial distribution of trees is discrete and independent variable data, their spatial distribution at the stand level, such as diameter at breast height and tree height, is directly affected by different continuous variables, such as light conditions, soil characteristics, temperature, and rainfall, so it can be assumed that these variables are continuous and spatially correlated [42]. Some scholars believe that spatial effects consist of spatial heterogeneity and spatial autocorrelation. Ignoring spatial effects in the modeling process may result in biased tests and suboptimal predictive models [43]. The spatial regression model can incorporate spatial effects into the regression model without requiring independent data [38,44]. Among the most commonly used models are the spatial lag model (SLM), the spatial error model (SEM), and the spatial Durbin model (SDM), which aim to capture the spatial dependence and autocorrelation of data [45–47]. However, when dealing with spatial heterogeneity, the geographically weighted regression (GWR) model is the preferred approach. GWR allows for a local analysis of the relationship between variables, providing more accurate and detailed results in situations where the relationship between variables differs spatially [48]. The use of remote sensing to determine



the spatial distribution of forest AGC and the challenge of reducing estimation uncertainty have gained widespread attention, especially the prevalent issues of overestimation and underestimation [49,50]. When the forest cover on the ground is too dense to be accurately distinguished by remote sensing methods, a data saturation phenomenon may occur.

This study focuses on the mixed coniferous forest AGC of a 2967 $km^2$ area in Fenglin County, Yichun City, Heilongjiang Province, Northeast China. It aims to offer a reference for estimating forest AGC in the Northeastern Forest Region. Using Sentinel-2 remote sensing images and 138 field-measured plots, both nonspatial and spatial regression models were employed to evaluate their capabilities in estimating forest AGC. The study contents and methods are described as follows: (1) we calculate the forest AGC storage using field measurements and remote sensing imagery processing and subsequently select modeling variables through stepwise regression analysis; (2) we construct ordinary least squares models and spatial regression models and compare their predictive abilities; (3) we analyze the inversion and spatial distribution of forest AGC in the study area; and (4) finally, we calculate AGC light saturation values resulting from data saturation phenomena during remote sensing estimation. The objective of this study is to use remote sensing technology to accurately estimate the spatial distribution of above-ground carbon storage in temperate forests, offering guidance and direction for forest dynamic monitoring. Additionally, it explores how various spatial regression models can enhance the estimation accuracy of forest AGC and alleviate challenges posed by data saturation.

## 2. Materials and Methods

### 2.1. Study Area

The study area is in Fenglin County, Yichun City, Heilongjiang Province, Northeast China, and includes the three forestry bureaus of Wuying, Hongxing, and Xinqing. It is situated on the southern slope of the Xiaoxing'an Mountains and serves as a quintessential example of the Northeastern Forest Zone. The Northeastern Forest Zone is an ecologically significant region in China, playing a pivotal role in timber production, biodiversity conservation, and maintaining environmental stability. The specific geographical location is shown in Figure 1. This area has a north temperate continental humid monsoon climate, with an annual average temperature of 0.6 °C and an annual precipitation of 500–610 mm [51]. Most of the precipitation is concentrated in June–August, which accounts for 73% of the total annual precipitation. The annual temperature variation range of the study site spans from −34 °C to 33 °C, and the average annual sunshine duration is approximately 2196 h. Historical meteorological data were sourced from the China National Meteorological Information Center (http://data.cma.cn/ (accessed on 15 April 2023)). This area is a natural forest, and the main vegetation types are mixed coniferous and broad-leaved forests, deciduous broad-leaved forests, and temperate coniferous forests. The dominant tree species include *Pinus*, *Betula*, *Larix*, *Picea*, *Abies*, *Fraxinus*, and *Populus*. This region has abundant diverse forest resources and is a national-level nature reserve. It contains the most representative forest type in Northeast China—coniferous and broad-leaved mixed forest dominated by Korean pine.

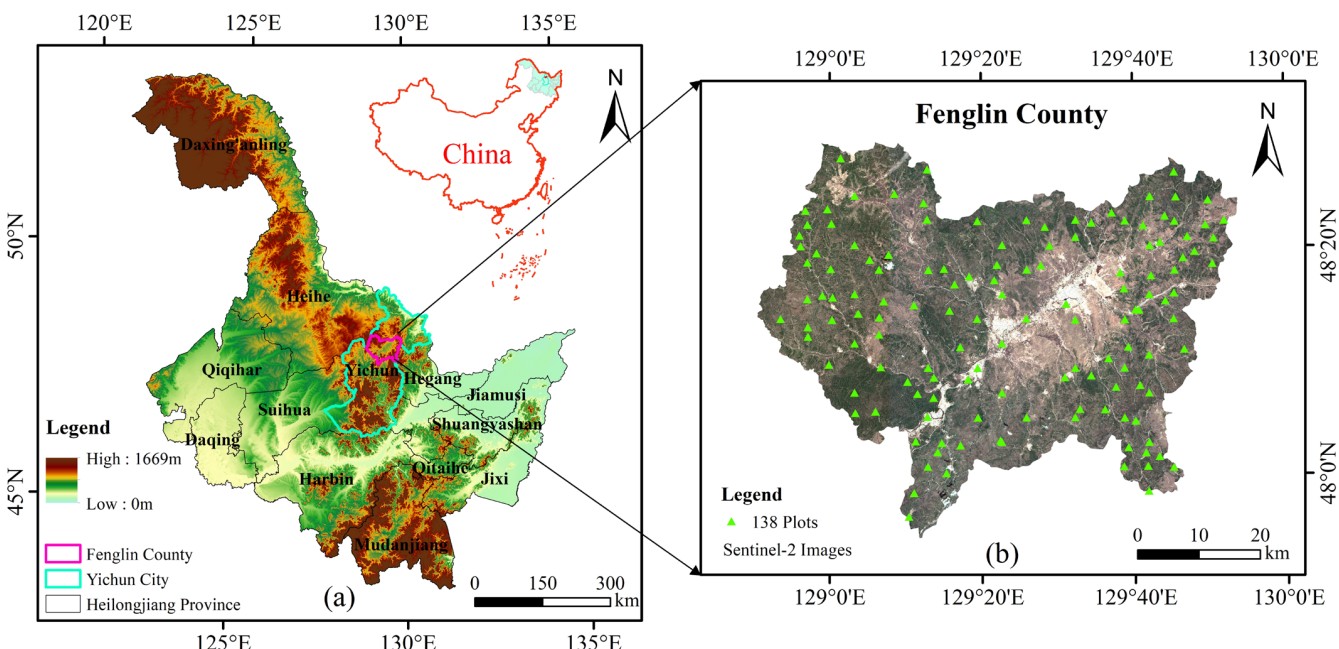

**Figure 1.** (**a**) The figure shows the location of the study area in Fenglin County, Heilongjiang Province, China, and the distribution of the digital elevation model (DEM) across Heilongjiang Province. (**b**) The figure shows the Sentinel-2 image of Fenglin County and the distribution of 138 plots in the study area in 2021.

### 2.2. Data Acquisition and Treatment

#### 2.2.1. Processing Standard Ground Survey Data

Standard ground survey data were sourced from 138 fixed distribution sample plots for forest fire risk investigation in Fenglin County in 2021. Each sample plot had an area of 0.06 ha, which translates to 600 m² or a square that is approximately 20 m × 30 m. The specific distribution locations of the sample plots are shown in Figure 1b.

The survey of the sample plot comprises various factors, such as the average diameter at breast height, average tree height, number of trees, forest age, vegetation type, and dominant tree species. Specifically, the DBH of the tree was measured using diameter tape, and the tree height and crown base height were measured with an ultrasonic altimeter. In addition, we also measured the location within the plot and the geographic coordinates of all trees using real-time kinematic (RTK) technology and Global Navigation Satellite System (GNSS) receivers. We use GNSS receivers to obtain the coordinates of the center point and four corner points of the sample plots. An example of a sample plot is shown in Figure 2.

The calculation of forest above-ground biomass (AGB) followed prior research on forest carbon storage in the Xiaoxing'an Mountains [52]. The AGB of individual trees for different tree species was calculated using the individual tree biomass model from the DBH data collected from each tree measured, as shown in Equations (1)–(3). The calculated values of $W_S$, $W_B$, and $W_L$ were multiplied by the corresponding carbon content coefficients provided in Table 1 to obtain $C_S$, $C_B$, and $C_L$ respectively. Finally, Equation (4) was used to calculate the AGC of a single tree. The sum of the AGC of individual trees in each plot divided by the unit area is the AGC of each plot, and the unit used is Mg/ha.

$$W_S = C_0 * D^{b_0} / \left(1 + r_2 * D^{k_2} + r_3 * D^{k_3} + r_4 * D^{k_4}\right) \tag{1}$$

$$W_B = r_2 * C_0 * D^{(k_2+b_0)} / \left(1 + r_2 * D^{k_2} + r_3 * D^{k_3} + r_4 * D^{k_4}\right) \tag{2}$$

$$W_L = r_3 * C_0 * D^{(k_3+b_0)} / \left(1 + r_2 * D^{k_2} + r_3 * D^{k_3} + r_4 * D^{k_4}\right) \quad (3)$$

where $W_S$ is the AGB of a single wood trunk; $W_B$ is the AGB of single wood branches; $W_L$ is the AGB of single wood leaves; $D$ is tree diameter at breast height; and $c_0$, $b_0$, $r_2$, $k_2$, $r_3$, $k_3$, $r_4$, and $k_4$ are the biomass model parameters of different tree species.

$$C = C_S + C_B + C_L \quad (4)$$

where $C$ is the AGC of a single tree; $C_S$ is the AGC of a single tree trunk; $C_B$ is the AGC of single tree branches; and $C_L$ is the AGC of single tree leaves.

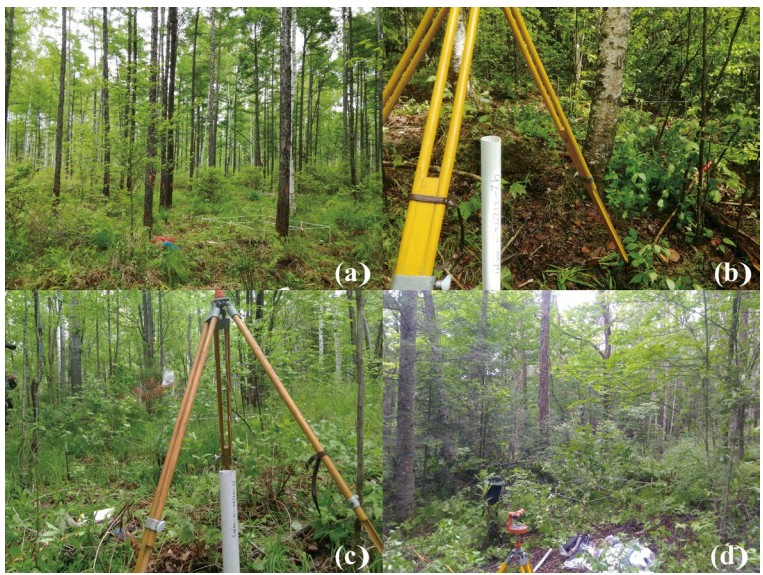

**Figure 2.** Sample plot photos. Figures (**a–d**) depict the site conditions of the actual study plots, with figures (**b**,**c**) specifically showing the standard plot corner stakes and the use of a compass clinometer for establishing the survey plot.

**Table 1.** Carbon storage conversion coefficients of different tree species in the Xiaoxing'an Mountains. (Num is the number of samples; SD is the standard deviation.)

| Species | Carbon Storage Conversion | | | Num | SD |
|---|---|---|---|---|---|
| | $C_S$ | $C_B$ | $C_L$ | | |
| *Picea koraiensis Nakai* | 0.4727 | 0.4875 | 0.4839 | 48 | 0.0407 |
| *Abies fabri (Mast.) Craib* | 0.4673 | 0.4783 | 0.5057 | 60 | 0.0406 |
| *Tilia amurensis Rupr.* | 0.4426 | 0.4255 | 0.4484 | 46 | 0.0212 |
| *Quercus mongolica* | 0.4558 | 0.4491 | 0.4672 | 64 | 0.0201 |
| *Ulmus pumila* | 0.4355 | 0.433 | 0.4322 | 40 | 0.0183 |
| *Acer pictum Thunb.* | 0.4422 | 0.4346 | 0.4462 | 46 | 0.0187 |
| *Betula dahurica Pall.* | 0.4529 | 0.4585 | 0.4639 | 52 | 0.0179 |
| *Betula platyphylla* | 0.4634 | 0.4619 | 0.4857 | 73 | 0.0229 |
| *Populus davidiana Dode* | 0.4430 | 0.4454 | 0.4587 | 54 | 0.0193 |
| *Pinus sylvestris var. mongolica* | 0.4775 | 0.4833 | 0.4967 | 85 | 0.0203 |
| *Pinus koraiensis Sieb. et Zucc* | 0.4807 | 0.4989 | 0.4924 | 34 | 0.0108 |
| *Larix gmelinii* | 0.4695 | 0.4761 | 0.4832 | 103 | 0.0311 |

### 2.2.2. Remote Sensing Data

Sentinel-2 is a large-scale, high-resolution multispectral imaging satellite funded by the European Union, the European Space Agency (ESA), and the Copernicus program. It is primarily used for monitoring the land environment and can provide observations on land vegetation growth, land cover status, and inland waterways and coastal areas.

The remote sensing data used in the study were Sentinel-2 Level-2A images acquired in April 2021, with image IDs N0300_R046_T52UEU and N0300_R046_T52UDU. To ensure data quality, the selected images over the study area were free of cloud cover. Level-2A images have undergone orthorectification, subpixel-level geometric accuracy correction, radiometric calibration, and atmospheric correction [53]. The remote sensing images are all downloaded from the Google Earth Engine platform (https://earthengine.google.com/platform/ (accessed on 2 March 2022)). The original remote sensing imagery consists of 12 bands. In this study, we only used bands with a resolution of 10 m, specifically band 2 (blue band: 0.458–0.523 μm), band 3 (green band: 0.543–0.578 μm), band 4 (red band: 0.650–0.680 μm), and band 8 (near-infrared band: 0.785–0.900 μm). In order to ensure that the pixel area of the remote sensing image matches that of the measured sample plot area, we employed the nearest neighbor interpolation method to resample the remote sensing image to a spatial resolution of 25 m. Subsequently, we used ENVI 5.3 software to crop the remote sensing image of the study area and to compute various remote sensing factors, including the original bands, vegetation indices, and texture features [54–58]. We computed the co-occurrence matrices for only the red, green, blue, and near-infrared bands, resulting in a total of 32 texture features. The corresponding formulas are shown in Table 2. Texture features derived from remote sensing imagery capture the visual homogeneity or heterogeneity within the image. These features represent specific variations in color or grayscale on the Earth's surface and are often indicative of the inherent properties of the surface objects [59–61]. Texture features provide image horizontal structure information and reflect the spatial variation in its gray values. When combined with vegetation indices, they can effectively depict the characteristics and changes in land features. In this study, terrain factors were extracted using a free digital elevation model (DEM). Downloaded from the Geospatial Data Cloud website (https://www.gscloud.cn/search (accessed on 19 March 2022)), ArcGIS 10.7 with the spatial and statistical analyst extensions was used to extract terrain factors, such as elevation, slope, and aspect.

**Table 2.** The calculation method of the vegetation index (B1, B2, B3, and B4 represent blue reflectivity, green reflectivity, red reflectivity, and near-infrared reflectivity, respectively).

| Type | Vegetation Index | Abbreviation | Calculation Formula |
|---|---|---|---|
| Original Band | B2-Blue | B1 | B2 |
| | B3-Green | B2 | B3 |
| | B4-Red | B3 | B4 |
| | B8-NIR | B4 | B8 |
| Vegetation Index | Ratio Vegetation Index | RVI | B8/B4 |
| | Atmospheric Ratio Vegetation Index | ARVI | $[B8 - (2 \times B4 - B2))/(B8 + (2 \times B4 - B2)]$ |
| | Soil Adjusted Vegetation Index | SAVI | $1.5 \times (B8 - B4)/8 \times (B8 + B4 + 0.5)$ |
| | Difference Vegetation Index | DVI | $B8 - B4$ |
| | Normalized Difference Vegetation Index | NDVI | $(B8 - B4)/(B8 + B4)$ |
| | Weighted Difference Vegetation Index | WDVI | $B8 - 0.5 \times B4$ |
| | Infrared Percentage Vegetation Index | IPVI | $B8/(B8 + B4)$ |
| | Red–Green Vegetation Index | RGVI | $(B4 - B3)/(B4 + B3)$ |
| | Triangular Vegetation Index | TVI | $0.5 \times [120 \times (B8 - B3)] - 200 \times (B4 - B3)$ |
| | Visible Atmospheric Resistance Index | VARI | $(B3 - B4)/(B3 + B4 - B2)$ |
| Texture | Mean | ME | $\sum_{i=0}^{N-1} \sum_{j=0}^{N-1} iP(i,j)$ |
| | Variance | VA | $\sum_{i=0}^{N-1} \sum_{j=0}^{N-1} (i - mean)^2 P(i,j)$ |
| | Homogeneity | HO | $\sum_{i=0}^{N-1} \sum_{j=0}^{N-1} \frac{P(i,j)}{1+(i-j)^2}$ |

**Table 2.** *Cont.*

| Type | Vegetation Index | Abbreviation | Calculation Formula |
|---|---|---|---|
| | Dissimilarity | CO | $\sum_{|i-j|=0}^{N-1}|i-j|^2\left\{\sum_{i=0}^{N}\sum_{j=0}^{N}P(i,j)\right\}$ |
| | Contrast | DI | $\sum_{|i-j|=0}^{N-1}|i-j|\left\{\sum_{i=0}^{N}\sum_{j=0}^{N}P(i,j)\right\}$ |
| Texture | Entropy | EN | $-\sum_{i=0}^{N-1}\sum_{j=0}^{N-1}P(i,j)log(P(i,j))$ |
| | Angular Second Moment | ASM | $\sum_{i=0}^{N-1}\sum_{j=0}^{N-1}P(i,j)^2$ |
| | Correlation | COR | $\frac{\sum_{i=0}^{N-1}\sum_{j=0}^{N-1}P(i,j)^2-\mu_x\mu_y}{\sigma_x\sigma_y}$ |

The Kolmogorov–Smirnov (K-S) test was used to check the normality of the research data. Then, the Pearson correlation was applied to study the relationship between AGC and factors such as forest stand, terrain, and remote sensing in the study area. Multiple stepwise regression analysis (MSR) was used to select the dependent variables for AGC modeling, with the significance levels for variable entry and removal set at 0.1 and 0.05, respectively. Variables were determined for collinearity based on a standard of variance inflation factor (VIF) of less than 10. Finally, utilizing the selected remote sensing variables through stepwise regression, we performed AGC estimation and spatial distribution analysis.

*2.3. Model Building and AGC Estimation*

2.3.1. Ordinary Least Squares Model

The ordinary least squares (OLS) model is used to obtain a best-fit model by incorporating data and prior information [62]. The dependent variable $Y$ is the AGC, with $n$ observations and $p$ independent variables $X$, such as forest, terrain, and remote sensing factors. The relationship between the independent variable $X$ and the dependent variable $Y$ can be expressed using linear regression, as shown in Equation (5):

$$Y = X\beta + \varepsilon \tag{5}$$

where $\beta$ is the model parameter and $\varepsilon$ is the model residual, which are assumed to follow a distribution. The parameters are estimated using the least sum of squares of the deviations between the dependent variable and the predicted values.

The OLS model is based on assumptions that apply to a whole region and is a global model where the constant and coefficients of explanatory variables are assumed to be the same across different study areas. However, the OLS model does not account for spatial autocorrelation and spatial heterogeneity between different regions.

2.3.2. Spatial Lag Model

The spatial lag model (SLM), also known as the spatial autoregressive model, is an autoregressive model that account for spatial variables [63]. When the dependent variable has significant spatial dependence on a spatial point, the spatial lag item can be introduced as a new explanatory variable in the classical statistical regression model. Assuming that the AGC of a sampling site is influenced by surrounding sampling sites, each sampling site can be viewed as a lagged effect of other sampling sites [64]. The SLM model is realized by adding the spatial lag item of the dependent variable $y$ to the OLS model, as shown in Equation (6):

$$Y = X\beta + \rho W_y + \varepsilon \tag{6}$$

where $\beta$ is the prognostic parameter; $W$ is the row-normalized spatial weight matrix; $W_y$ is the weighted average of adjacent sample sites; $y$ is the spatial lag item; $\rho$ is the spatial autocorrelation parameter, which is influenced by the matrix $W$; and $\varepsilon$ is a random error item that obeys an $N(0, \sigma^2 I)$ normal distribution.

### 2.3.3. Spatial Error Model

The spatial error model (SEM) refers to a model where the error term is spatially correlated, meaning that the spatial correlation is attributed to the error term rather than the systematic part of the model [65]. The SEM assumes that the spatial autocorrelation is considered from the error term without changing the explanatory variables, thereby estimating the spatial autocorrelation coefficient. Specifically, the model error is partitioned into two components: the error caused by spatial autocorrelation and the error from the model itself, as shown in Equation (7).

$$Y = X\beta + \lambda W_\varepsilon + \xi \tag{7}$$

where $\lambda$ is the spatial autocorrelation parameter; $W_\varepsilon$ is the spatial error term; and $\xi$ is a random error item that obeys an $N(0, \sigma^2 I)$ normal distribution.

### 2.3.4. Spatial Durbin Model

The spatial Durbin model (SDM) is an extended form that combines the SLM and SEM by incorporating corresponding constraints on these models [66]. This model considers the spatial autocorrelation of both the dependent and independent variables and can be formulated as shown in Equation (8).

$$Y = \rho WY + X\beta + \lambda \rho WX\beta + \varepsilon \tag{8}$$

where $\rho$ is the spatial autoregressive coefficient, which indicates the strength of spatial dependence. $W$ is the spatial weight matrix, which describes the degree of spatial interdependence in the sample. $\lambda$ is the spatial lag coefficient, and $\rho WY$ and $\lambda \rho WX\beta$ denote the spatially lagged dependent and independent variables, respectively.

### 2.3.5. Geographically Weighted Regression Model

The geographically weighted regression model (GWR) is widely recognized as one of the most effective methods for addressing spatial heterogeneity [44,67]. This model extends the global regression model by building a regression model at each point in space, weighting all observations using a distance function from nearby points [68]. The aim is to identify spatial patterns by estimating a set of coefficient values for each point by moving a window over the data [69]. The basic form of the model is shown in Equation (9).

$$Y_{(u_i, v_i)} = \beta_{0(u_i,v_i)} + \beta_{1(u_i,v_i)} X_{1i} + \beta_{2(u_i,v_i)} X_{2i} + \cdots + \beta_{n(u_i,v_i)} X_{ni} + \varepsilon_i \tag{9}$$

where $(u_i, v_i)$ is the coordinate at point $i$; $Y_{(u_i,v_i)}$ is the dependent variable at point $i$; $n$ is the number of samples; $X_{ni}$ is the value of the $n$th variable at point $i$; $\beta_0$ is the intercept; and $\varepsilon_i$ is the error term. In this model, the parameters of each sampling point are estimated based on the spatial weight matrix ($W_i$), where $W_i$ is a diagonal matrix of spatial weights for point $i$, and $W_i = f(d_i, h)$, where $d_i$ is the distance vector between location $i$ and all neighbors and $h$ is the bandwidth. We used an adaptive bisquare kernel function to select the optimal bandwidth, enabling the detection of nonstationary relationships that global models might overlook.

### 2.3.6. Model Accuracy Evaluation Method

The sample plots are divided into 103 training data sets and 35 testing data sets by random sampling, and RStudio 4.2.1 is employed to fit OLS, SEM, SLM, SDM, and GWR models. The model fitting accuracy is commonly evaluated using the correlation coefficient ($R^2$), the adjusted coefficient of determination ($R^2_{adj}$), the relative root mean squared error (rRMSE), the mean absolute percentage error (MARE), the mean absolute error (MAE), and the mean percentage bias (MPE) [70]. Typically, a higher $R^2$ and $R^2_{adj}$, as well as a lower

rRMSE, MARE, MAE, and MPE, indicate better performance of the model. These statistical analyses are expressed in Equations (10)–(15).

$$R^2 = 1 - \frac{\sum_{i=1}^{n}(y - \hat{y}_i)^2}{\sum_{i=1}^{n}(y - \overline{y}_i)^2} \tag{10}$$

$$R_{adj}^2 = 1 - \frac{(1 - R^2)(n - 1)}{n - k - 1} \tag{11}$$

$$\text{rRMSE} = \sqrt{\frac{\frac{1}{n}\sum_{i=1}^{n}(y - \hat{y}_i)^2}{\overline{y}}} \times 100\% \tag{12}$$

$$\text{MARE} = \frac{1}{n}\sum_{i=1}^{n}\frac{|y - \hat{y}_i|}{y_i} \tag{13}$$

$$\text{MAE} = \frac{1}{n}\sum_{i=1}^{n}|y - \hat{y}_i| \tag{14}$$

$$\text{MPE} = \frac{\sum_{i=1}^{n}|y - \hat{y}_i|}{\sum_{i=1}^{n}y_i} \times 100\% \tag{15}$$

In this study, we analyzed spatial autocorrelation and heterogeneity in forest AGC and model residuals using global and local Moran's I methods [71,72], as shown in Equation (16). A positive Moran's I indicates similar residual levels, while a negative value suggests contrasting trends. If Moran's I is near zero, the residuals are randomly distributed with no mutual influence [73,74]. Moran's I was used to measure the global autocorrelation between the AGC and model prediction residuals.

$$\text{I} = \frac{n\sum_{i=1}^{n}\sum_{j=1}^{n}w_{ij}(d)(x_i - \overline{x})(x_j - \overline{x})}{\sum_{i=1}^{n}\sum_{j=1}^{n}w_{ij}(d)\sum_{i=1}^{n}(x_i - \overline{x})^2} \tag{16}$$

where $n$ is the sample size, $x_i$ is the observed values at different locations, $\overline{x}$ is the average of the observed values at different locations, and $w_{ij}(d)$ is the weight based on the distance between sample points $i$ and $j$. The Z value is a multiple of the standard deviation, and the significance of Moran's I is tested by the Z value to determine whether the spatial autocorrelation of the observed values exists. The Z value calculation method is shown in Equation (17).

$$\text{Z(I)} = \frac{\text{I} - \text{E(I)}}{\sqrt{\text{Var(I)}}} \tag{17}$$

If the Z value falls within the range of −1.96 to +1.96, the uncorrected *p* value will be greater than 0.05. Therefore, the null hypothesis cannot be rejected, indicating that the spatial distribution of AGC is likely to be random and does not exhibit any spatial effects. If the Z value falls outside this range, the spatial pattern displayed would be a statistically significant cluster or dispersion trend [75]. Lastly, we used the local spatial autocorrelation tool in ArcGIS 10.7 to visualize the spatial cluster or dispersion trend [76].

### 2.3.7. Confirmation of Light Saturation Value

Optical remote sensing images can capture the unique spectral characteristics of different vegetation types. When the forest cover on the ground is too dense to be accurately distinguished by remote sensing methods, a data saturation phenomenon may occur. Since different remote sensing images and different estimation methods will produce different saturation values, this study defines the saturation value of light as a range, indicating that when the AGC reaches this range, using remote sensing images for AGC estimation will result in saturation. Previous studies on the saturation value of above-ground carbon storage often used the most relevant variables to build nonlinear or spherical models and

calculated the extreme values as the saturation value of above-ground carbon storage [11]. However, using a single variable to estimate the saturation of above-ground carbon storage may result in the loss of valuable information, leading to low and inaccurate precision. The generalized additive model (GAM) is an additive modeling technique in which the predictor variables are modeled by a smoothing function [77]. This approach is highly flexible and offers strong interpretability. The quantile generalized additive model (QGAM) is based on the GAM, employing quantile regression for predictions. Compared to traditional methods, this approach can use more variables and avoid information loss. This study attempts to estimate the AGC saturation value using linear, quadratic, and logarithmic functions and GAM and QGAM. QGAM is fitted using seven quantile points (q = 0.2, 0.3, 0.4, 0.5, 0.6, 0.7, 0.8) to estimate the AGC saturation value in Fenglin County using range values to replace single-point values and eliminate estimation errors caused by maximum and minimum values. Unlike traditional regression models that only focus on the mean, QGAMs allow us to model in different distribution regions of the data, thus gaining a more comprehensive understanding of the nature of the data. The GAM and QGAM are fitted using functions from the gam and qgam packages in R [78] Figure 3 shows a flowchart of the steps used in our study.

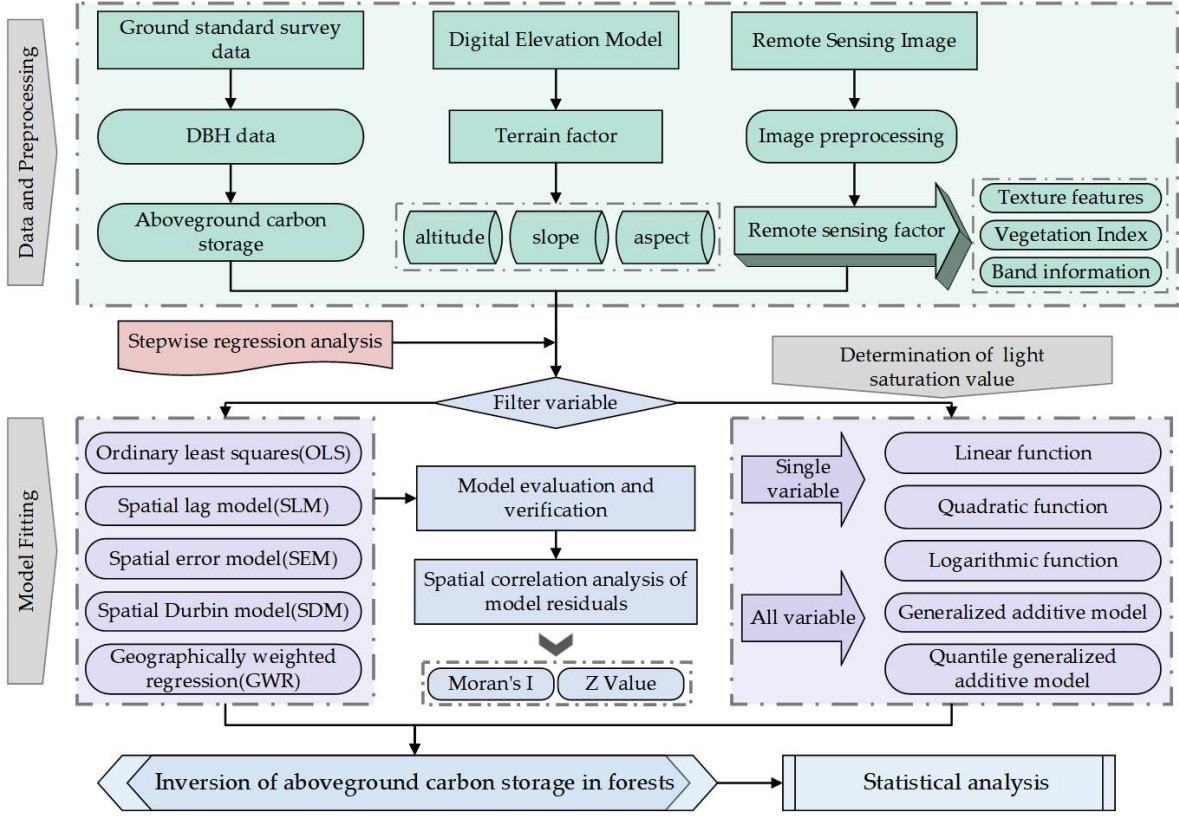

**Figure 3.** Flowchart of steps used in our study. The research is divided into several parts, including basic data processing, model fitting, determination of light saturation value, AGC spatial distribution inversion, and statistical analysis.

## 3. Results

### 3.1. Variable Screening

Descriptive statistics for the data are shown in Table 3. Based on the K-S test results, as the null hypothesis assumes no difference between the data and a normal distribution, and with a *p*-value significantly greater than 0.05, we cannot reject the null hypothesis, indicating that the data can be considered to follow a normal distribution. According to the correlation analysis between AGC and remote sensing factors shown in Figure 4,

26 remote sensing factors exhibit a significant correlation at the $p < 0.05$ level. A stepwise regression method was used for optimal variable selection, and finally, four remote sensing variables, IPVI, B3EN, SLOPE, and Aspect, were selected to estimate and analyze the spatial distribution of AGC in Fenglin County. The mean AGC measured in the sample plot is 63.121 Mg/ha, which is attributed to the natural secondary forest being in the middle-aged stage. Due to the presence of a few trees with a diameter at breast height greater than 0.3 m in some plots, the AGC in those areas exhibits higher values, where peak value reaches up to 153.058 Mg/ha. All VIF values are below 10, indicating the absence of multicollinearity among the selected variables. This aids in achieving a stable, interpretable, and accurate model outcome, thereby preventing overfitting.

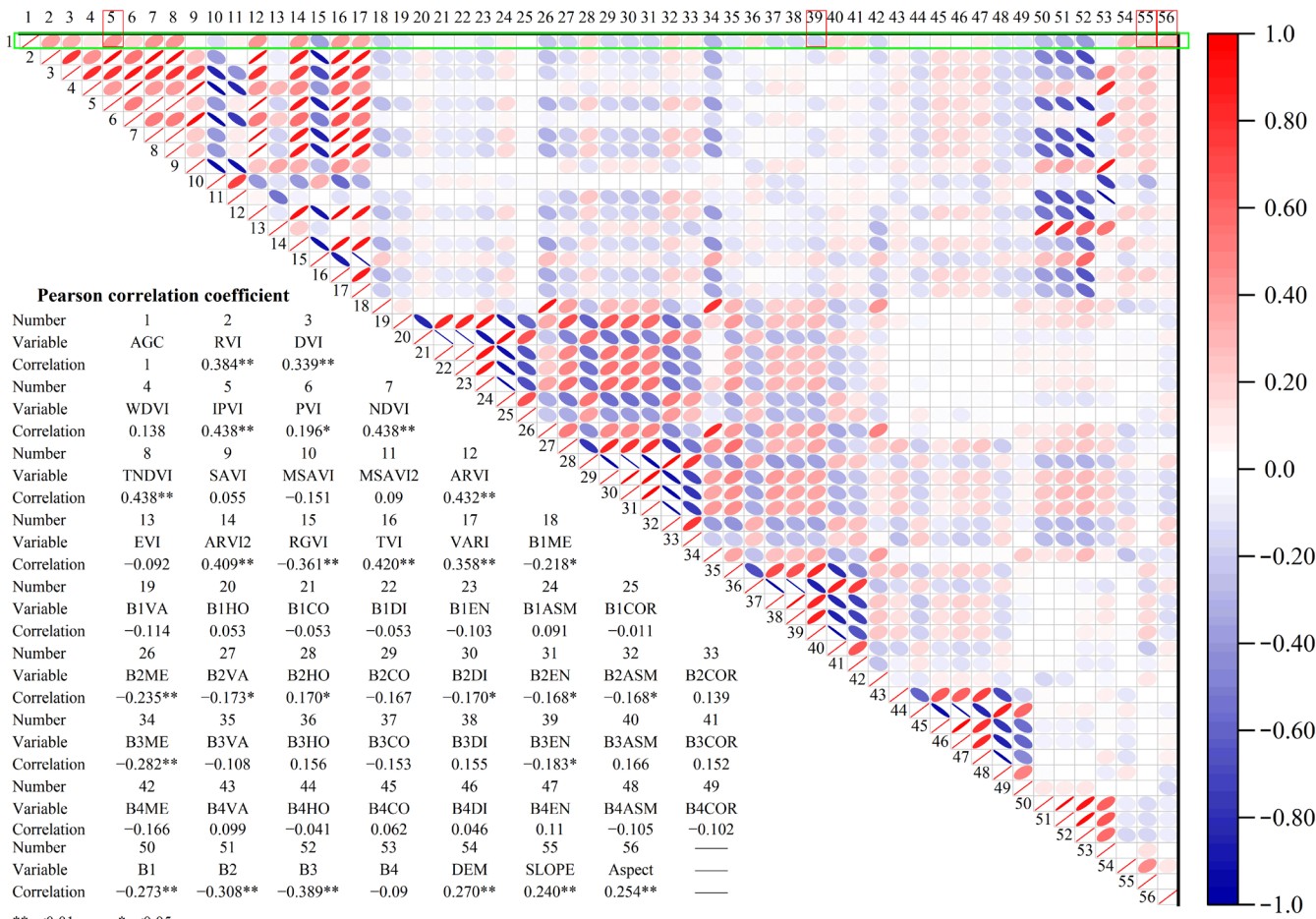

**Pearson correlation coefficient**

| Number | 1 | 2 | 3 | | | |
|---|---|---|---|---|---|---|
| Variable | AGC | RVI | DVI | | | |
| Correlation | 1 | 0.384** | 0.339** | | | |
| Number | 4 | 5 | 6 | 7 | | |
| Variable | WDVI | IPVI | PVI | NDVI | | |
| Correlation | 0.138 | 0.438** | 0.196* | 0.438** | | |
| Number | 8 | 9 | 10 | 11 | 12 | |
| Variable | TNDVI | SAVI | MSAVI | MSAVI2 | ARVI | |
| Correlation | 0.438** | 0.055 | −0.151 | 0.09 | 0.432** | |
| Number | 13 | 14 | 15 | 16 | 17 | 18 |
| Variable | EVI | ARVI2 | RGVI | TVI | VARI | B1ME |
| Correlation | −0.092 | 0.409** | −0.361** | 0.420** | 0.358** | −0.218* |
| Number | 19 | 20 | 21 | 22 | 23 | 24 | 25 |
| Variable | B1VA | B1HO | B1CO | B1DI | B1EN | B1ASM | B1COR |
| Correlation | −0.114 | 0.053 | −0.053 | −0.053 | −0.103 | 0.091 | −0.011 |
| Number | 26 | 27 | 28 | 29 | 30 | 31 | 32 | 33 |
| Variable | B2ME | B2VA | B2HO | B2CO | B2DI | B2EN | B2ASM | B2COR |
| Correlation | −0.235** | −0.173* | 0.170* | −0.167 | −0.170* | −0.168* | −0.168* | 0.139 |
| Number | 34 | 35 | 36 | 37 | 38 | 39 | 40 | 41 |
| Variable | B3ME | B3VA | B3HO | B3CO | B3DI | B3EN | B3ASM | B3COR |
| Correlation | −0.282** | −0.108 | 0.156 | −0.153 | 0.155 | −0.183* | 0.166 | 0.152 |
| Number | 42 | 43 | 44 | 45 | 46 | 47 | 48 | 49 |
| Variable | B4ME | B4VA | B4HO | B4CO | B4DI | B4EN | B4ASM | B4COR |
| Correlation | −0.166 | 0.099 | −0.041 | 0.062 | 0.046 | 0.11 | −0.105 | −0.102 |
| Number | 50 | 51 | 52 | 53 | 54 | 55 | 56 | |
| Variable | B1 | B2 | B3 | B4 | DEM | SLOPE | Aspect | —— |
| Correlation | −0.273** | −0.308** | −0.389** | −0.09 | 0.270** | 0.240** | 0.254** | —— |

**p<0.01   *p<0.05

**Figure 4.** Correlation analysis of AGC and remote sensing factors. The value range of the Pearson correlation coefficient is between −1 and 1, red ($p > 0$) represents a positive correlation, and blue ($p < 0$) represents a negative correlation. The smaller and darker the ellipse is, the higher the correlation between the two variables. The green box represents the correlation between the dependent variable AGC and each independent variable. The correlation coefficients are listed in the bottom-left corner of the table. The variables selected as the final choices for stepwise regression are surrounded by red boxes: 5, 39, 55, and 56.

In the field of geography and spatial data analysis, the instability of the relationship between spatially distributed variables is called spatial nonstationarity. To study the spatial nonstationarity of different variables and AGC, a scatter diagram of AGC and different variables in space is shown in Figure 5. The distribution of AGC is lower in the central part of the study area, and each variable shows different trends as latitude and longitude increase. The distribution of IPVI closely mirrors that of AGC, both exhibiting a trend

where values are elevated at the periphery of the study area while diminishing towards the central region.

**Table 3.** Descriptive statistics of independent variables and remote sensing variables. D: Kolmogorov–Smirnov distance; P: Kolmogorov–Smirnov test *p*-value.

| Variable | Num | Min | Median | Max | Mean | Unit | Std | VIF | D | *p* |
|---|---|---|---|---|---|---|---|---|---|---|
| AGC | 138 | 6.130 | 57.375 | 153.058 | 63.121 | Mg/ha | 30.951 | — | 0.090 | 0.213 |
| IPVI | 138 | 0.582 | 0.774 | 0.895 | 0.769 | — | 0.067 | 1.074 | 0.091 | 0.207 |
| B3EN | 138 | 0.000 | 0.349 | 1.581 | 0.502 | — | 0.472 | 1.087 | 0.108 | 0.964 |
| SLOPE | 138 | 0.155 | 4.787 | 15.133 | 5.494 | ° | 3.570 | 1.098 | 0.118 | 0.927 |
| Aspect | 138 | 0.000 | 168.368 | 347.005 | 170.557 | ° | 96.123 | 1.022 | 0.176 | 0.571 |

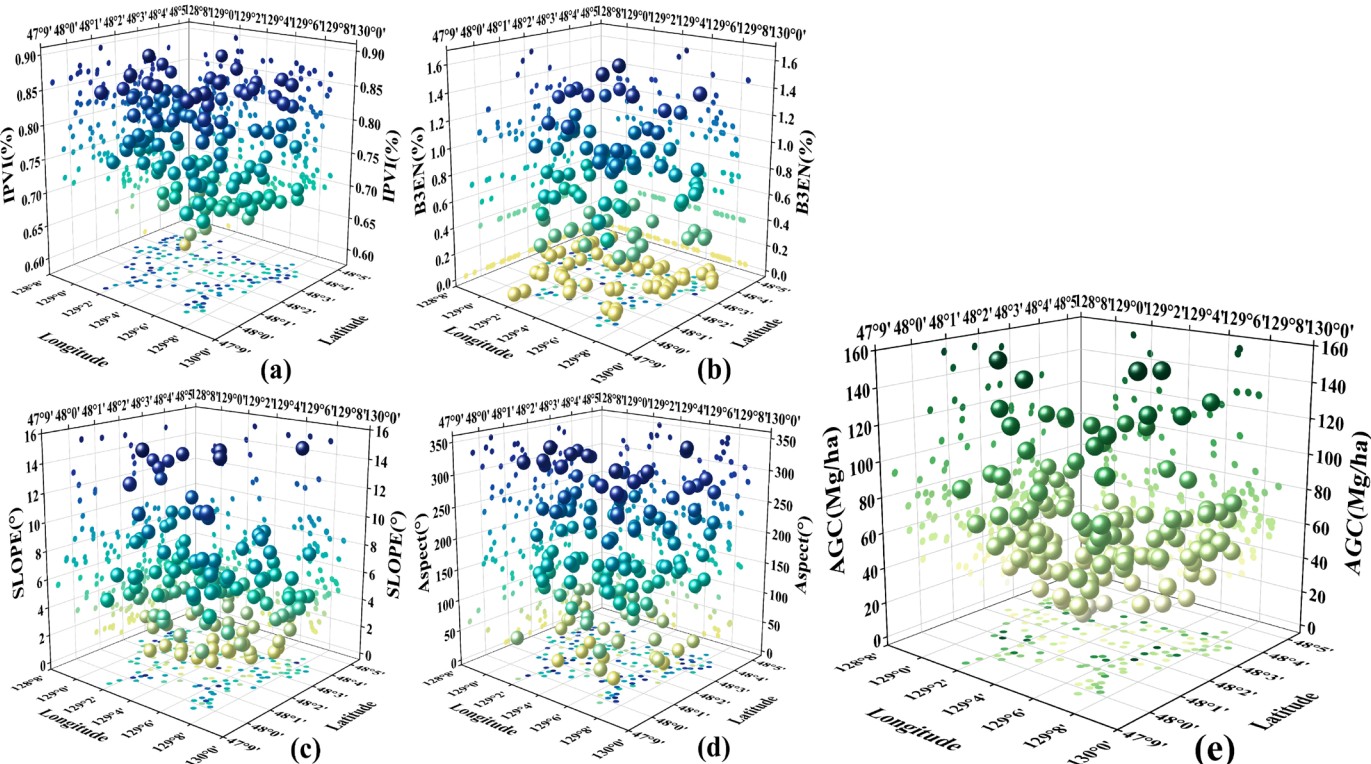

**Figure 5.** Variation trend of different variables and actual above-ground carbon reserves along longitude and latitude. The X-axis represents longitude, the Y-axis represents latitude, and the Z-axis represents the values of different variables. Figures (**a**–**e**) illustrate the spatial distribution of variables IPVI, B3EN, SLOPE, Aspect, and AGC, showing their respective values at different locations.

## 3.2. Spatial Correlation Analysis

The independent variables selected after the stepwise regression were standardized, and Moran's I test of spatial correlation was carried out on OLS. The results in Table 4 show that the *p*-value is close to zero, indicating that there is significant spatial autocorrelation in the residuals of the OLS model. Therefore, when constructing the AGC model of Fenglin County, it is necessary to consider the spatial effect and solve the AGC estimation error caused by the spatial effect.

**Table 4.** Moran's I test for spatial correlation in residuals.

| Moran's I | Moran's I Statistic | Marginal Probability | Mean | Standard Deviation |
|---|---|---|---|---|
| 0.8590 | 53.0058 | 0.0000 | −0.0004 | 0.0162 |

Using local spatial correlation analysis to explore the spatial distribution of various types of clusters, Figure 6 shows the cluster distribution of AGC. Spatial correlation refers to the phenomenon where objects that are close to each other spatially tend to have similar trends and values in their attributes. Conversely, if objects that are spatially close have different trends and values in their attributes, this spatial correlation manifests as a negative spatial correlation, which is characterized by the presence of a "high–high cluster" or "high–low outlier" distribution [79,80]. Table 5 shows the Z-score statistics for local Moran's I. In the study area, there is one sample plot showing a low–high outlier, possibly due to its proximity to a road. There are three plots of land that show high–low outliers, which account for 2.174% of the total sample. Additionally, there are eight plots of land that exhibit high–high clustering, representing 5.797% of the total sample and showing a clear positive correlation. Finally, there are eleven plots of land that display low–low clustering, which account for 7.971% of the total sample.

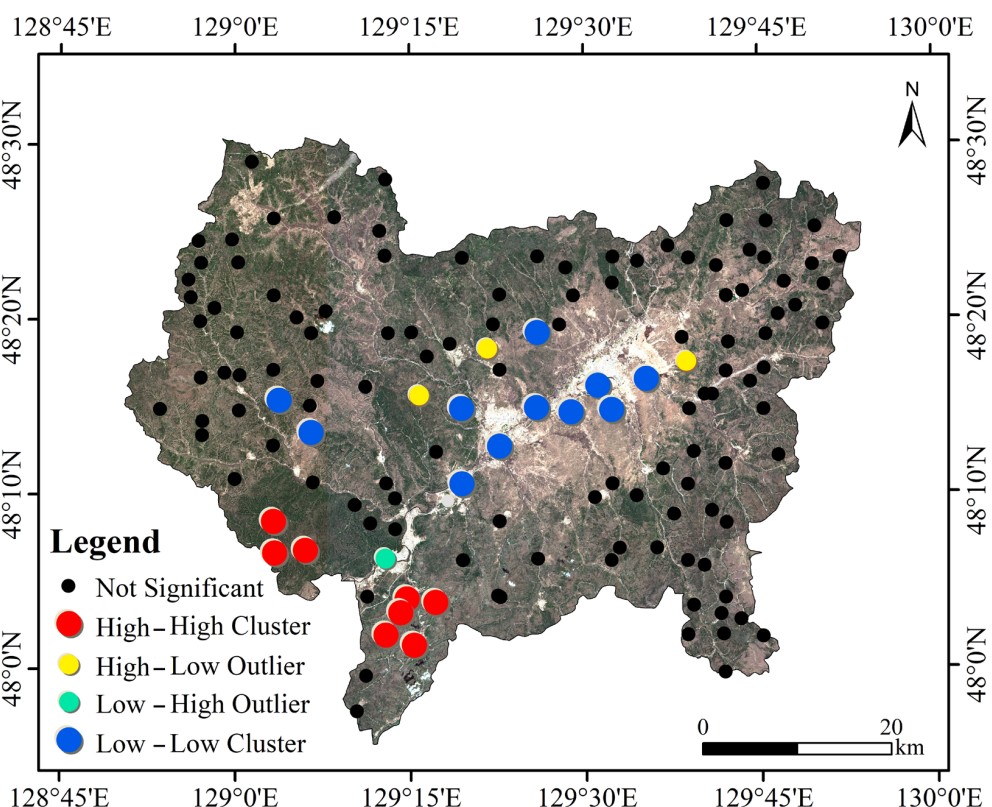

**Figure 6.** Local spatial autocorrelation. The red points represent high–high clusters; the yellow points represent high–low outliers; the green points represent low–high outliers; the blue points represent low–low clusters; and the black points represent nonsignificant local spatial autocorrelation.

**Table 5.** Z score statistics of local Moran's I. LH: low–high outlier; HL: high–low outlier; HH: high–high cluster; LL: low–low cluster.

| Z Score | Type | Number | Percentage |
|---|---|---|---|
| <−2.58 | LH | 1 | 0.725% |
| −2.58~−1.96 | HL | 3 | 2.174% |
| −1.96~−1.65 | — | 0 | 0.000% |
| −1.65~1.65 | — | 115 | 83.333% |
| 1.65~1.96 | HH | 8 | 5.797% |
| 1.96~2.58 | LL | 10 | 7.246% |
| >2.58 | LL | 1 | 0.725% |

### 3.3. AGC Model Evaluation

The accuracy evaluation and prediction effect of each model are shown in Table 6. The fitting effect and predictive ability of the OLS model are poor, and the SDM model performs best in the global regression model, which indicates that the SDM model can better fit the global structure of the data. The GWR model has the highest $R^2$ (0.695) and the smallest rRMSE (27.329), MARE (0.280), MAE (14.858), and MPB (22.734) values. To further visualize the model fitting performance, Figure 7 shows scatter plots of the observed and predicted AGC for 103 plots based on the OLS model and 4 spatial regression models. The global regression models exhibit a clear tendency to overestimate at low values and underestimate at high values of AGC when it is less than 40 Mg/ha or greater than 100 Mg/ha. The GWR estimation threshold of forest AGC was expanded from 0–100 Mg/ha to 0–120 Mg/ha. The GWR model not only offers superior predictive accuracy but also outperforms the other four global regression models in both fitting and predictive performance.

**Table 6.** Comparison of modeling results.

| Models | Training Set (*n* = 103) | | | | | | Validation Set (*n* = 35) | | | |
| | $R^2$ | $R^2_{adj}$ | rRMSE (%) | MARE (Mg/ha) | MAE (Mg/ha) | MPB (%) | rRMSE (%) | MARE (Mg/ha) | MAE (Mg/ha) | MPB (%) |
|---|---|---|---|---|---|---|---|---|---|---|
| OLS | 0.320 | 0.299 | 40.813 | 0.385 | 21.367 | 32.694 | 41.167 | 0.544 | 18.384 | 32.509 |
| SLM | 0.326 | 0.306 | 40.623 | 0.385 | 21.302 | 32.594 | 41.035 | 0.543 | 18.381 | 32.504 |
| SEM | 0.327 | 0.306 | 40.617 | 0.385 | 21.346 | 32.662 | 40.983 | 0.540 | 18.337 | 32.427 |
| SDM | 0.371 | 0.352 | 39.251 | 0.379 | 20.591 | 31.506 | 39.184 | 0.522 | 18.407 | 32.551 |
| GWR | 0.695 | 0.686 | 27.329 | 0.280 | 14.858 | 22.734 | 28.927 | 0.394 | 13.908 | 24.595 |

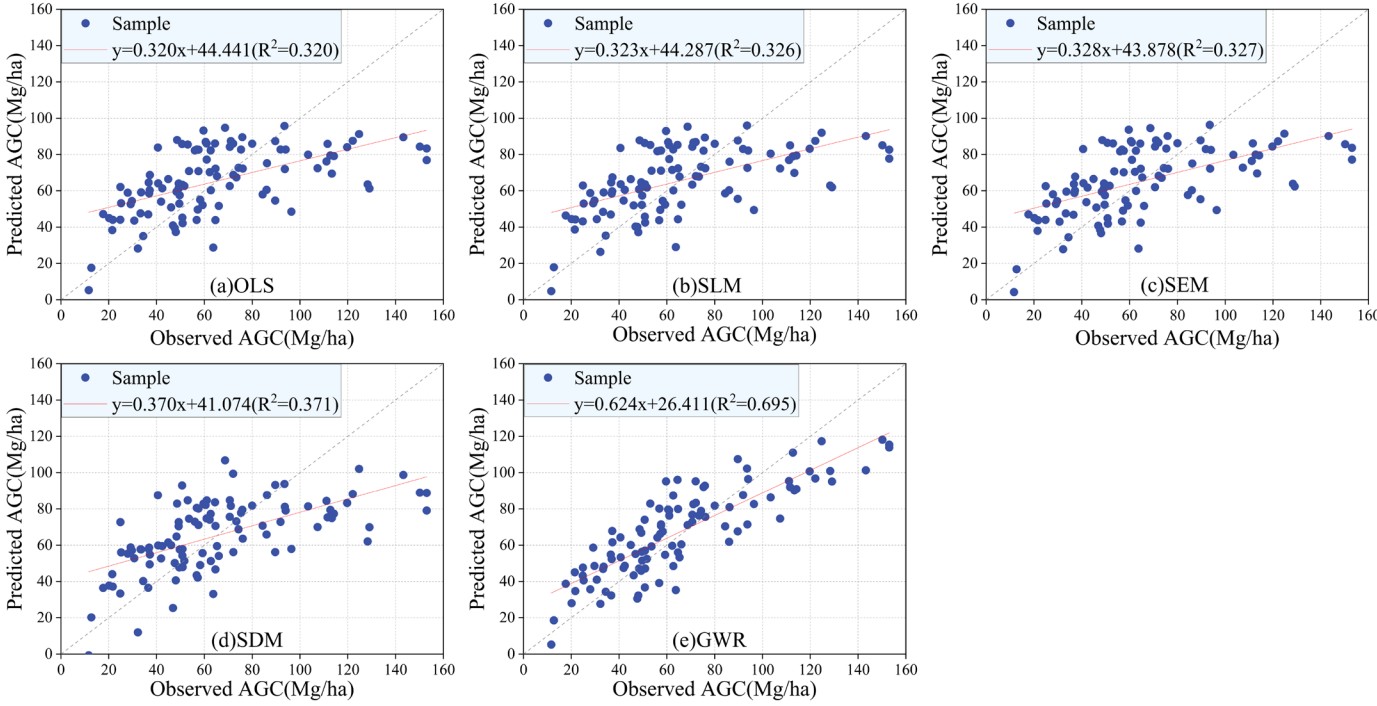

**Figure 7.** The relationship between observed and predicted AGC (Mg/ha) for 103 plots using the ordinary least squares model (OLS) and 4 spatial regression models (SLM, SEM, SDM, and GWR). The blue points represent the sample data, the dashed line represents the central line, and the red line represents the fitted line. The closer the red line is to the central line, the better the model fit is.

Furthermore, according to the variance analysis results of the GWR model shown in Table 7, the GWR model has improved compared to the OLS model. The sum of squared residuals decreased by 33,306.914, indicating a decrease in the overall variation in the

model residuals. The degrees of freedom also decreased by 40.442, and the mean squared residuals decreased by 823.571. These findings suggest that the OLS model had significant spatial autocorrelation and spatial heterogeneity in the residuals, while the GWR model could address the spatial effects present in the model residuals.

Figure 8 shows the spatial distribution of AGC in Fenglin County based on five different model inversions. This observation suggests a lower distribution of AGC in the central region, contrasting with a higher distribution in the southwestern region. This finding is consistent with the results of the high–high clustering discussed in Section 3.2. The OLS model tends to overestimate low values and underestimate high ones. While global regression improves estimation slightly, the GWR model significantly enhances the accuracy for both high and low value areas, aligning more closely with the true AGC distribution. The AGC distribution in Fenglin County exhibits a pattern of gradual increase from the sparser central region to the outer periphery, with areas of 100–120 Mg/ha encompassing 10.40% and those surpassing 120 Mg/ha occupying 3.99% of the total area. By using remote sensing information extracted from Sentinel-2 images and estimating the distribution of AGC in Fenglin County through a spatial regression model, the results obtained are consistent with the actual situation, providing a reference for analyzing the spatial distribution of forest AGC using remote sensing.

**Table 7.** Variance analysis of the GWR model. Sum Sq: sum of squares of mean deviations; DF: degree freedom; Mean Sq: mean square; F: value of F.

| Source | Sum Sq | DF | Mean Sq | F |
|---|---|---|---|---|
| OLS Model Residuals | 73,278.508 | 98.000 | — | — |
| GWR Model Improvement | 33,306.914 | 40.442 | 823.571 | — |
| GWR Model Residuals | 39,971.593 | 57.558 | 694.458 | 1.186 |

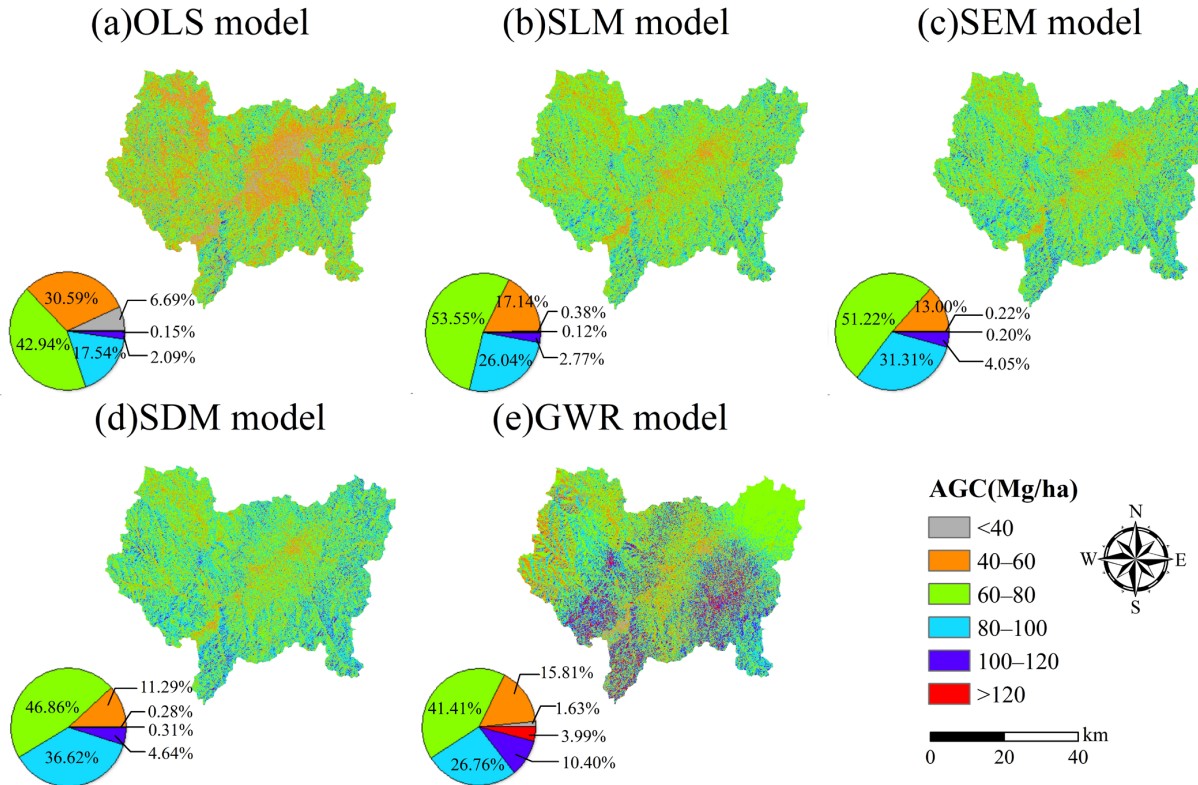

**Figure 8.** Spatial distribution of forest AGC in Fenglin County by five models in 2021: (**a**) OLS, (**b**) SLM, (**c**) SEM, (**d**) SDM, and (**e**) GWR model. The pie chart shows the proportion of carbon storage area distribution in each interval.

### 3.4. Spatial Correlation Analysis

The residual Moran's I of five models for nine different bandwidths in the range of 0 m to 9000 m are compared, as shown in Figure 9. As the bandwidth increases, Moran's I gradually approaches zero, indicating that the spatial autocorrelation of the model residuals also decreases as the spatial scale increases.

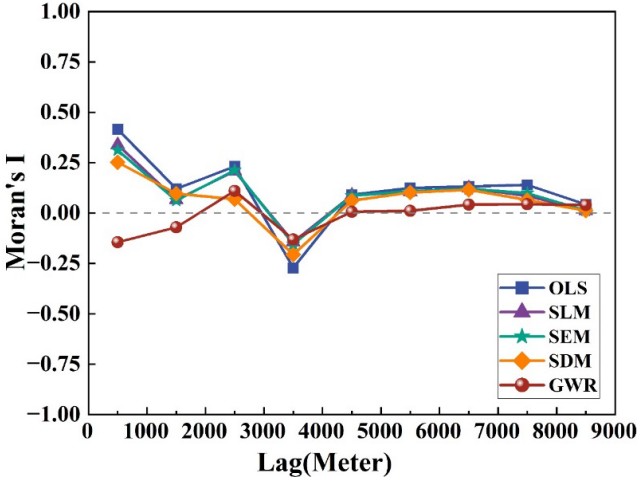

**Figure 9.** Moran's I of the model residual under different bandwidths.

The five models' residual Moran's I and the corresponding Z values are listed in Table 8. The Z values of OLS and SLM are significantly positive (Z value > 1.96), indicating that these two models are similarly clustered at a significance level of 0.05, while GWR is negative with an absolute value below 1.96. The LISA cluster map provides a clear visualization of the spatial distribution of HH, LL, LH, and HL regions. Figure 10 displays the four significant autocorrelations present within the study area. From the figure, it is evident that the low–low clusters are primarily located in the central region, while the high–high outliers are predominantly situated in the southeastern and southwestern regions. Overall, the GWR has significantly reduced the impact of spatial autocorrelation.

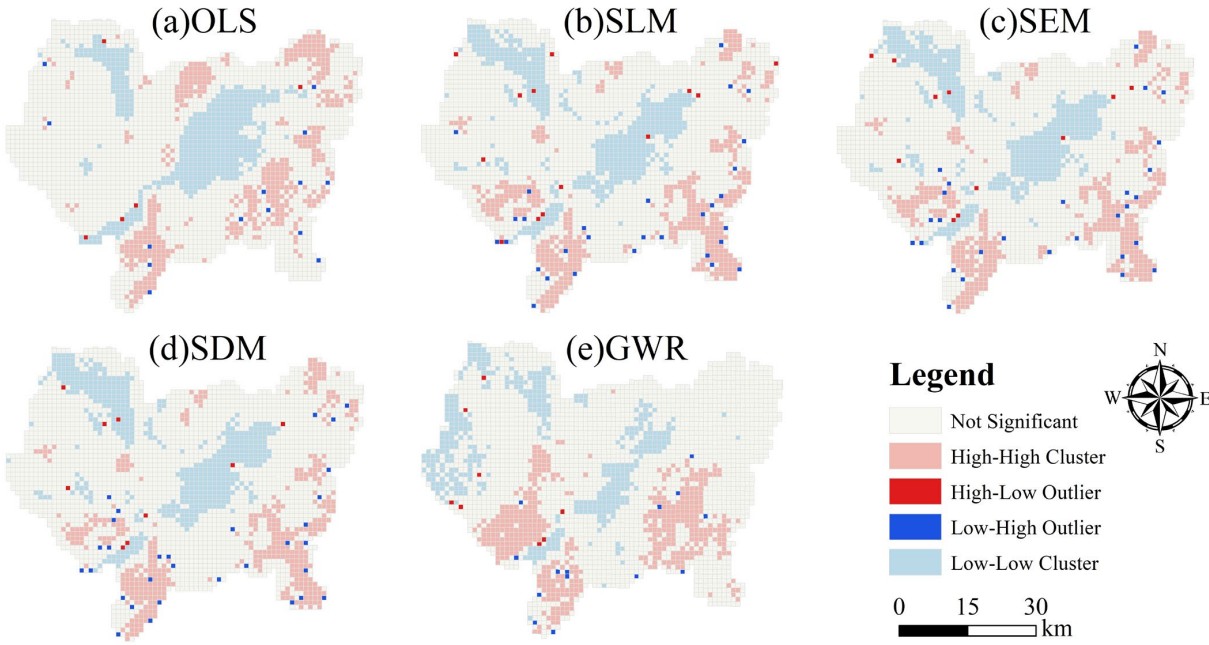

**Figure 10.** LISA cluster map of the five models.

**Table 8.** Moran's I and Z score values for the prediction residuals of the five models.

| Model | Moran's I | Z Value |
|---|---|---|
| OLS | 0.415 | 2.602 |
| SLM | 0.338 | 1.967 |
| SEM | 0.312 | 1.805 |
| SDM | 0.252 | 1.413 |
| GWR | −0.145 | −0.565 |

### 3.5. Determination of Light Saturation Value

Table 9 shows the fitting estimation results of the AGC light saturation value using the linear regression and nonlinear regression models. The maximum value of the regression results represents the estimated light saturation value of AGC under the respective method. It can be seen that all variables are more interpretable than a single variable, with higher $R^2_{adj}$ and explained deviance and smaller AIC values. Previous studies on optical saturation often relied on a single variable for prediction. This study demonstrates that appropriately additional predictive variables can improve the model's fitting accuracy, resulting in more realistic and reliable predictions. Although the GAM offers the best fit, it is one-sided to represent the light saturation value of AGC by a single value. The results of fitting all variables to the QGAM can be used as a reference for the range of AGC saturation values.

**Table 9.** Estimation results of the AGC light saturation value in Fenglin County.

| Variable | Function | | $R^2_{adj}$ | AIC | Max | DE |
|---|---|---|---|---|---|---|
| IPVI | Linear function | | 0.198 | 1312.48 | 89.608 | — |
| | Quadratic function | | 0.216 | 1310.4 | 76.925 | — |
| | Logarithmic function | | 0.163 | 1315.36 | 91.706 | — |
| | GAM | | 0.217 | 1310.15 | 82.975 | 22.80% |
| | QGAM | 0.2 | 0.2 | 1278.44 | 54.544 | 60.80% |
| | | 0.3 | 0.206 | 1286.69 | 59.864 | 44.40% |
| | | 0.4 | 0.208 | 1292.72 | 65.058 | 26.30% |
| | | 0.5 | 0.213 | 1305.99 | 71.932 | 17.10% |
| | | 0.6 | 0.218 | 1328.18 | 83.863 | 15.10% |
| | | 0.7 | 0.213 | 1345.05 | 99.007 | 21.70% |
| | | 0.8 | 0.184 | 1377.54 | 117.145 | 37.60% |
| All variables | Linear function | | 0.285 | 1299.67 | 100.763 | — |
| | GAM | | 0.294 | 1298.8 | 97.071 | 31.90% |
| | QGAM | 0.2 | 0.271 | 1264.83 | 65.48 | 64.70% |
| | | 0.3 | 0.279 | 1278.12 | 71.285 | 52.90% |
| | | 0.4 | 0.274 | 1284.29 | 77.862 | 36.80% |
| | | 0.5 | 0.274 | 1294.29 | 85.026 | 27.40% |
| | | 0.6 | 0.279 | 1310.25 | 95.702 | 23.90% |
| | | 0.7 | 0.285 | 1331.79 | 108.832 | 28.40% |
| | | 0.8 | 0.273 | 1397.39 | 129.894 | 43.50% |

To further visualize the QGAM, this study shows the QGAM with 138 plot fits applied to the most significant variable IPVI, as shown in Figure 11. In both low-value and high-value regions, the model demonstrates exemplary fitting performance. Relative to a singular linear model, this approach markedly elevates the predictive accuracy, adaptability, and interpretability. At the same time, it allows for segmented evaluation of AGC prediction outcomes from different spatial regression models.

Based on the light saturation value statistics of a single variable and all variables based on QGAM, the proportions of forest AGC area in Fenglin County inverted by each spatial regression model are shown in Figure 12. The figure reveals that the saturation phenomenon is more prominent in the QGAM results based on all variables, and significant differences in estimation among the models are evident within the q range of 0.7–0.8.

We consider the results obtained from the QGAM constructed using all variables, within the q value range of 0.7 to 0.8, as the range for AGC light saturation values (108.832 to 129.894 Mg/ha). The global regression model fails to accurately estimate forest AGC within the saturation range and is also incapable of predicting AGC values exceeding this range. The inversion results from the GWR model reveal that the forest AGC in Fenglin County is saturated in approximately 6.26% of the total area and exceeds saturation in about 1.97% of the area. The GWR model to some extent addresses the issue of data saturation when utilizing remote sensing for AGC estimation.

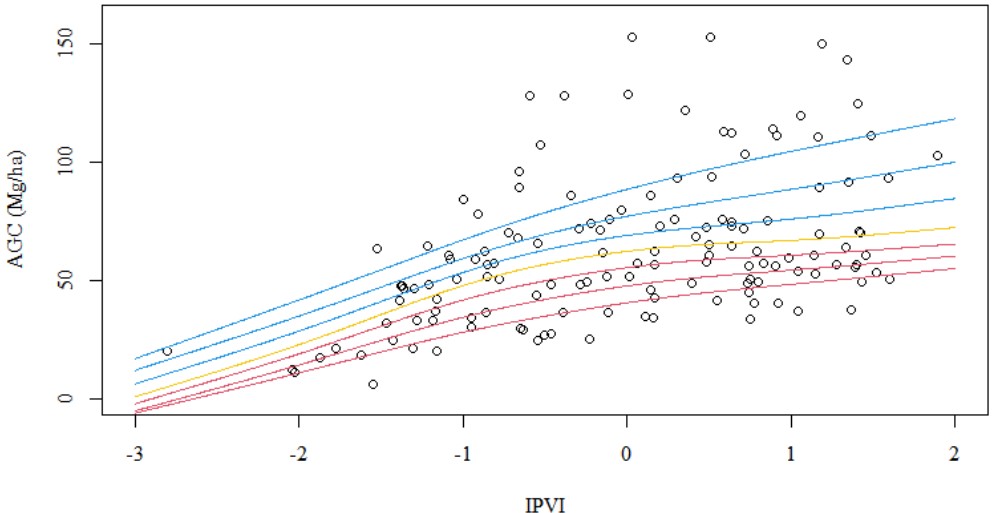

**Figure 11.** The QGAM fitted by the most significant variable IPVI (after standardization) (note: the red line is the QGAM of q = 0.2, 0.3, 0.4 from bottom to top; the yellow line is the QGAM of q = 0.5; the blue line is from bottom to top QGAM with q = 0.6, 0.7, 0.8, respectively). The white circle is the AGC data of the measured sample plot.

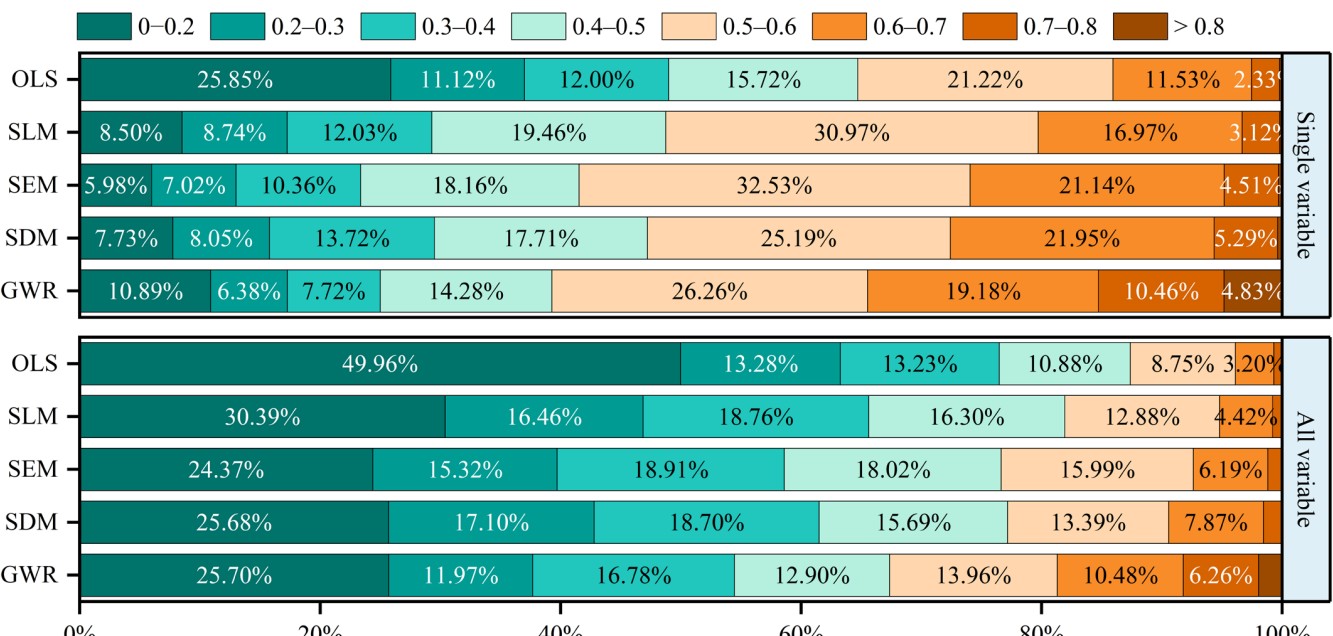

**Figure 12.** Based on the light saturation value results of QGAM's single variable and all variables, the statistics of the proportion of forest AGC area in Fenglin County across each spatial regression model's inversions.

## 4. Discussion

### 4.1. Uncertainty Analysis of AGC Estimation

Uncertainties in forest AGC estimation using remote sensing primarily stem from the selection of AGC modeling variables and the choice of algorithm for constructing the AGC estimation model [26]. In this study, a spatial regression model was used to estimate the AGC. From the above model fitting and prediction results, the spatial effect cannot be ignored. Spatial effects can be caused by many factors, such as distance-related species interactions, spatial nonstationarity among variables, and nonlinear relationships between environmental factors and species that are erroneously modeled as linear [81]. Notably, some sample plots in the study area are distributed near villages within the study area, exhibiting lower AGC values. Simultaneously, there is a large spatial heterogeneity among the plots due to the influence of altitude. In this study, GWR was used to explore the spatial distribution of large-scale sample plots. By considering spatial correlation, the AGC spatial heterogeneity is reduced for model construction, which has a strong improvement compared with the OLS model fitting prediction. The GWR model reduces uncertainty in remote sensing estimates by accounting for spatial heterogeneity and correlation, thus mitigating overestimation in high-value areas and underestimation in low-value areas [82]. This finding is consistent with the results obtained by Ou et al., who used Landsat 8 and a spatial regression model for predicting AGB [38]. For AGC estimation methods, there are also nonparametric and machine-learning models, which are higher than GWR in terms of fitting accuracy [30,83]. Li et al. employed Sentinel-2 and used four machine learning methods to estimate forest AGC in Shanghai. They discovered that the model yielding the best predictive results still exhibited instances of overestimation. The researchers attributed this phenomenon to the uneven distribution of samples and the presence of significant spatial heterogeneity within the sample data [30]. Puliti et al. improved the estimation accuracy of AGB in Norway by utilizing ArcticDEM and Sentinel-2 data in conjunction with a random forest model. They indicated that forest characteristics and terrain are sources of uncertainty in model predictions [84]. Although the nonparametric method closely responds to sample data characteristics, its sensitivity is accentuated given the small sample size in this study. In contrast, GWR not only offers superior predictive capabilities but also delivers a more lucid mathematical interpretation. It emphasizes the spatial distribution of the studied multivariate relationship and adeptly accounts for the influences of spatial autocorrelation and heterogeneity on local-scale AGC estimation, making it well suited for analyzing spatial nonstationarity in dynamic environments [85].

### 4.2. Light Saturation Phenomenon

The issue of light saturation is common when estimating AGC using remote sensing data [10]. In a previous study, Steininger used Landsat data to determine the age and aboveground biomass of 34 tropical secondary forest sites in Manaus, Brazil, and found that data saturation occurred when the above-ground biomass approached approximately $15 \text{ kg/m}^2$ or when the vegetation reached an age of 15 years or more and canopy reflectance was saturated [86]. Mature forest stands with complex structures can also cause data saturation, and there are many reasons why this may occur [10,87]. Ahmad et al. conducted a study on biomass estimation in moist temperate forests in the Galies region of Abbottabad, Pakistan, using Sentinel-2 remote sensing data. They discovered that the accuracy of Sentinel-2-derived indices was influenced in areas with higher vegetation density [88]. To improve the reliability of above-ground biomass carbon estimation, integrating data from multiple sensors, stratifying AGC estimation based on vegetation types and slope, and combining age virtual variables and texture features have been proposed [26,89]. Visible light saturation is an important factor that results in inaccuracies in estimations of high AGC values. Researchers studying the biomass saturation of temperate forests in China have obtained saturation values that are not significantly different from the results of this study [11]. In this study, the QGAM method was used to accurately estimate the forest AGC light saturation value from 108.832 to 129.894 Mg/ha. It was found that more than 8% of

the forest AGC in Fenglin County inverted by the GWR model was in the saturation range, which improved the problem of data saturation compared with the global regression model. This finding indicates that the forest AGC area falling within the light saturation value is not small and the uncertainty caused by data saturation issues cannot be overlooked. Accurately estimating the saturation value of forest AGC is crucial for formulating sensible management strategies and environmental protection policies.

*4.3. Limitations and Future Works*

This study highlights the benefits of using remote sensing for forest AGC estimation. However, it should be noted that the image quality of optical remote sensing is often compromised by cloud cover. It is recommended to select remote sensing images from spring or summer with cloud cover less than 2% for processing and analysis in order to enhance the usability of the image data. The dataset used in this study is uniformly distributed and representative. The fixed distribution plots measured in the field ensure high data quality. This allows us to not only provide a more comprehensive and accurate analysis of the spatial distribution of forest AGC in Fenglin County but also offer a reference for its precise estimation in the Northeast Forest Region. Additionally, Sentinel-2 data and spatial regression models can also be used for spatial distribution analysis in other fields, such as the spatial variation in crops, soil organic carbon distribution, and air quality [28,46,58]. In the future, it will be essential to analyze the spatial distribution of carbon storage in other forest types and even broader ecosystem contexts. With the implementation of forest conservation policies, it is anticipated that future forest distribution will be dominated by mature forests. It is imperative to delve deeply into the data saturation challenges encountered in remote sensing estimation.

**5. Conclusions**

Based on Sentinel-2 remote sensing images, we constructed a spatial regression model to predict the spatial distribution of cold temperate forest AGC in Northeast China. This study resulted in the following conclusions: (1) The GWR model constructed by combining vegetation index texture features and terrain factors has the best fitting accuracy and predictive performance. It shows the highest $R^2$ (0.695) and the lowest rRMSE (0.273). Following closely were the SDM, SEM, SLM, and OLS models in terms of their performance. (2) The spatial effect should not be ignored, as evidenced by the analysis of model residuals and the spatial distribution inversion of AGC. (3) During AGC estimation using optical remote sensing, a saturation phenomenon occurs. The AGC light saturation values estimated through QGAM range from 108.832 to 129.894 Mg/ha, with a saturated area percentage of 8.23% for forest AGC in Fenglin County.

**Author Contributions:** Conceptualization, S.W. and Y.S.; methodology, S.W.; software, S.W.; validation, Y.S.; formal analysis, S.W.; investigation, F.W.; resources, Y.S.; data curation, Y.S.; writing—original draft preparation, S.W.; writing—review and editing, Y.S., H.Z. and S.L.; visualization, S.W.; supervision, S.W.; project administration, W.J.; funding acquisition, W.J. All authors have read and agreed to the published version of the manuscript.

**Funding:** This research was funded by the China National Key Research and Development Program (Grant No.2022YFD2201003-02) and the Special Fund Project for Basic Research in Central Universities (2572019CP08, 2572022DT03).

**Data Availability Statement:** Data are contained within the article.

**Conflicts of Interest:** The authors declare no conflict of interest.

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
