# Peer review of "Estimation of Above-Ground Carbon Storage and Light Saturation Value in Northeastern China’s Natural Forests Using Different Spatial Regression Models"

_forests, doi:10.3390/f14101970_

Round 1

Reviewer 1 Report

Dear Authors,

    Your manuscript "Estimation of Above-ground Carbon Storage and Light Saturation Value in Northeastern China's Natural Forests Using Different Spatial Regression Models" presents a relevant but well-explored topic. I have read your article thoroughly and have provided minor comments in the comments of the digital archive. Below I present comments that require more attention:

1) Abstract: This needs to be revised as it is confusing, especially between lines 27 to 33.

2) Line 86: "LiDAR data enable the acquisition of canopy height information, are not limited by cloud coverage". In fact, the wavelength in the NIR or Green band interacts with cloud droplets and therefore Lidar is interfered with by cloud cover. Revise.

3) Line 161: Please revise the values referring to the temperature variation of the study site: 1900 to 2200° C?

4) Figure 2: Explain the meaning of the letters.

5) Revise the text presented in lines 189 to 195. It looks like a Journal recommendation.

6) Line 226: What method was used to resample the images?

7) Please quote the unit used to calculate the slope;

8) Line 227: Please detail the spatial registration step. What is its purpose? Level 2 images are orthorectified and do not require a registration step.

9) Lines 228 and 229: explain the texture step. Was the co-occurrence matrix actually calculated for each band?

10) Table 2: Use the same band notation as for the Sentinel 2 MSI sensor: B2-blue, B3-Gree, B4-Red, and B8-NIR

11) Line 243: "The Pearson correlation coefficient was used to analyze the correlation". In this case, do the variables have a normal distribution for the Pearson coefficient to be used?

12) Correlation analysis alone tends not to be very explanatory. It would be interesting in this case to use PCA or even RFE (Recursive Feature Elimination), which is more robust and tends to provide a more accurate analysis (ranking of variables).

13) Was the data processed only in the ArcGIS application with the Spatial and Statistical Analyst extensions and the R application? Please quote the hardware configuration presented in the study.

14) Line 361: Comment on the processing flowchart shown in Figure 3.

15) Line 375: "There is no multicollinearity problem among the respective variables". Discuss this issue and its implications further.

16) Figure 4: The correlation values obtained are low. With weak correlation values like those obtained, how can we expect to obtain a good estimate of AGC? This result needs to be better explored and justified because statistically the results are weak.

17) Line 427: "The sample plots are divided into 103 training data sets and 35 testing data sets". Present this information in chapter 2.3.6

18) Lines 506 to 510: deepen the discussion as there is a very interesting result here.

19) Conclusions: Make recommendations for future studies.

   I end my review by congratulating you on your study and the version of the manuscript you have submitted.

Sincerely,

Be careful with punctuation, especially the use of commas. Proofread the text carefully, as a section of the text contains a recommendation from the journal. Pay attention to verb and time agreement. The text needs to be proofread by a native English speaker to improve it, for example, "of the surface of objects" for an object's surface.

Reviewer 2 Report

The manuscript addressed one of the important global research topics supported by a well-developed model. This manuscript has great potential to contribute to minimizing the uncertainties in carbon estimation, which is important for sustainable land use planning and policy formulation. There are some minor issues that can be considered for further improvement of the manuscript.

Suggestions: 
1. The introduction section seems too big! Please precise it by focusing on the general background, rationale, research gaps, and objectives. Moreover, this paper's scientific contribution or novelty should be more clearly stated. What issue do the authors want to answer in the study, for example? What makes this study different from others? This information should be included in the introduction.

2. Author could present the relevant maps of spatial autocorrelation in detail, i.e., Moran's I, LISA cluster, and LISA significance map, to better understand the hotspot and cold spot. Authors may follow the paper of Zhang et al., 2019. 

Zhang, F., Yushanjiang, A., & Jing, Y. (2019). Assessing and predicting changes of the ecosystem service values based on land use/cover change in Ebinur Lake Wetland National Nature Reserve, Xinjiang, China. Science of the Total Environment, 656, 1133-1144.

3. Author could also mention some of the limitations of the study that can be considered in further study.

4. In Conclusion, one paragraph should be based on main findings and the second paragraph should state the policy suggestions along with the future research directions in a good order. Present policy suggestions are not very sound. Please provide a sound policy implications stemming from the main findings of this work.

5. Indications of future work may be pointed out to guide interested readers to the subsequent work. 

Reviewer 3 Report

General comments:

The paper offers valuable insights into the complexities of AGC estimation using remote sensing data, particularly in cold temperate forests. The exploration of the light saturation phenomenon and its implications stands out as a significant contribution to the field. The research design and methodology employed are sound and robust. However, while the paper is systematically structured, some sections might benefit from improved clarity. Given the paper's significance and contribution to the field, I recommend acceptance after minor revisions.

Abstract

The abstract is dense with technical terms, making it overwhelming. Consider breaking up some of the sentences for increased clarity.

Be cautious about making generalized statements like "trees are subject to varying degrees of environmental influence." (lines 14–15). While true, such broad statement could benefit from further specifics in the main paper.

Ensure consistency in mentioning places. The abstract mentions the whole of Northeast China (line 19) but then specifies Fenglin County (line 28). Is Fenglin County representative of Northeast China, or is it just one case study? Clarifying this can prevent potential confusion.

It would be helpful to elaborate briefly on the implications of your research. For instance, how might your findings impact sustainable development strategies?

Introduction

Some statements are reiterated (e.g., the importance of sustainable development and significance of forests are mentioned more than once). You should aim for conciseness. The repeated emphasis on the importance of sustainable development or significance of forests can be merged into one concise statement.

While the challenges of AGC estimation are described, reinforcing the specific aspects of your study's approach can underscore the paper's significance. The introduction could benefit from some concise framing when discussing the importance of AGC and its ecological functions.

Giving a brief overview of the significance of studying Fenglin County might be useful. Why were this location chosen? Does it represent an exceptional challenge or opportunity in terms of AGC estimation?

I believe that “multisensor” data (line 82 and elsewhere), should be “multisensory” data.

A few sentences could be structured better to enhance readability. For example, "LiDAR data enable the acquisition of canopy height information, are not limited by cloud coverage and lighting conditions and offer the advantages of high-precision estimation." (lines 86–88). Rephrasing suggestion: "LiDAR data, unaffected by cloud coverage and lighting conditions, offer high-precision estimation by capturing canopy height information."

You may want to highlight how your study fills a gap in the current literature.

(Lines 41–42) "Forests are an important component of both resource issues and the environment." This sentence can be clarified. Are forests a solution to these issues, or are they part of the problem?

Materials and Methods

In Figure 1a, the outline of the study area is not clearly visible due to a color discrepancy. Additionally, there is an inconsistency between the coordinate data displayed above and below the figure. For Figure 1b, the sampling plots, represented by red triangles, are challenging to distinguish against the grayscale background due to a color contrast issue. While the legend references Sentinel-2 images (band 2, band 3, band 4), it is unclear how these bands are visually represented within the figure itself.

Figure 2 needs a detailed caption that explicitly explains the representations of a, b, c, and d.

There seems to be a placeholder paragraph from the template still embedded within your paper on lines 189–195.

A brief explanation of why specific bands from Sentinel-2 were chosen can help readers understand the reasoning behind these decisions.

The decision to resample the 10 m*10 m Sentinel-2 image to 25 m*25 m requires justification (lines 226–227). Why was this done? How does this change affect the data's interpretation?

How did you account for potential errors or inconsistencies in the Sentinel-2 remote sensing data?

Why is the consideration of the Pearson correlation coefficient significant for AGC analysis?

The basis of the OLS model is clearly outlined (Section 2.3.1). However, explicitly mentioning the limitations of this model in handling spatial autocorrelation would be beneficial for context.

The SLM explanation is understandable (Section 2.3.2). However, a diagram might aid in better conceptualization of how neighboring sampling sites influence the AGC of your particular site.

The SDM explanation combines features of both SLM and SEM (Section 2.3.4). It might be helpful to provide a comparative summary of these three models for better clarity.

The section 2.3.5 does a commendable job of explaining how GWR addresses spatial heterogeneity. However, it would be beneficial to expand on the advantages/disadvantages of using adaptive bisquare kernel functions. The GWR model demonstrated the best performance in your study. Are there specific environmental or data factors that might make other models more suitable in different conditions or regions?

The inclusion of various statistical measures to evaluate model performance is thorough (Section 2.3.6). While the Z value's role is clear, offering real-world implications of the spatial cluster or dispersion trend can offer more intuitive insight.

The section 2.3.7 does a great job highlighting the potential pitfalls of AGC estimation using remote sensing and the importance of saturation value. However, it may be helpful to emphasize why GAM and QGAM were chosen over other potential methods. The paper mentions using gam and qgam packages in R. If feasible, providing extracts of the code or directing readers to a supplementary material with code can be of value to those looking to replicate or extend this research.

While each model's fundamentals are laid out, it would be beneficial to discuss the assumptions and limitations briefly for each. Offer a table or short summary comparing the models in terms of their applicability, strengths, weaknesses.

Results

Ensuring consistency in presenting results will improve clarity. For instance, in the statement "breast height of more than 30 cm," (line 375) the units were provided, but it would be beneficial to consistently provide such details throughout.

There is some repetition in the presentation. For example, the superiority of the GWR model is mentioned multiple times. While it is essential to emphasize key findings, it might be more effective to consolidate these mentions. How did you address potential outliers in the data, especially when presenting results using the GWR model?

In Figure 6, there appears to be a color overlap that could lead to misinterpretations. The colors used for the "high-high cluster" and "high-low outlier" are nearly indistinguishable, which can be misleading. Similarly, the "low-high outlier" seems to blend into the background. Utilizing gray circles for non-significant data against a gray background also compromises clarity. I would recommend choosing more distinct and contrasting colors to enhance the figure's readability and comprehension.

Discussion

Sentences are lengthy, making them hard to follow. Consider breaking them into shorter, more concise sentences.

There seems to be a mix of past and future perspectives (e.g., "In the future, we will explore..."). It would be beneficial to maintain consistent tense throughout or clearly demarcate sections discussing past findings from future research directions.

Consider including a concluding paragraph at the end of the discussion that encapsulates the key findings and their implications.

How do the results of this study potentially change the way AGC is estimated in Northeast China or other similar regions?

Could you elaborate on the real-world implications of these findings?

The light saturation values you found in Fenglin County are quite specific. Do you anticipate these values to vary significantly in other parts of Northeast China or in other temperate forest environments?

Could you elaborate on how the light saturation phenomenon manifests in other landscapes or geographical regions? Are there notable differences when compared to cold temperate forests?

Conclusions

There is some repetition, especially regarding the benefits of the GWR model. This can be condensed for brevity.

While the conclusions do a good job of summarizing the findings, they might benefit from a brief statement regarding potential implications or broader significance of your findings.

Consider expanding slightly on the potential applications or implications of your findings in real-world scenarios, such as forest conservation or climate change mitigation.

A brief mention of future research directions or potential limitations of the current study might provide a more rounded conclusion.

How do you envision the scalability of your models for larger regions or even at a global scale?

The paper is written in comprehensible English, conveying the research findings effectively. However, there are instances where sentence structure and phrasing could be enhanced for better clarity and flow. Additionally, there are a few areas where sentences could benefit from restructuring for clarity. Furthermore, simplifying complex sentences and avoiding overly lengthy explanations will enhance readability. Some minor grammatical and punctuation inconsistencies were noted, and comprehensive proofreading is recommended to address these discrepancies.

Reviewer 4 Report

Paper was good and presentation was also perfect. Few corrections are required before acceptance.

Abstract: Good

Introduction: Compactness is required, rest is ok 

Study area: Need Improvement (See the attached PDf)

Statistical analysis is missing

Results and Discussion: Good

Conclusion: perfect

References: Check in Ms and Reference section carefully

Round 2

Reviewer 1 Report

Dear Authors,

     The manuscript "Estimation of Above-ground Carbon Storage and Light Saturation Value in Northeastern China's Natural Forests Using Different Spatial Regression Models" in its second version has several changes that have made it easier to read and understand the experiments/results.

       With regard to my questions, I would just like to ask about the normality of the data: although you did not test for normality, you did analyze descriptive statistics and graphs. In this context, you have indirectly assessed the normality of the data and it would be interesting to point out that the data has a normal distribution.

        I conclude by congratulating you on the new version of the manuscript.

Sincerely,
